# Transcriptome Analysis of Roots from Wheat (*Triticum aestivum* L.) Varieties in Response to Drought Stress

**DOI:** 10.3390/ijms24087245

**Published:** 2023-04-14

**Authors:** Wei Xi, Chenyang Hao, Tian Li, Huajun Wang, Xueyong Zhang

**Affiliations:** 1College of Agronomy, Gansu Agricultural University, Lanzhou 730070, China; xiwei923700915@163.com; 2State Key Laboratory of Aridland Crop Science/Gansu Key Laboratory of Crop Improvement & Germplasm Enhancement, Gansu Agricultural University, Lanzhou 730070, China; 3Key Laboratory of Crop Gene Resources and Germplasm Enhancement, Ministry of Agriculture and Rural Affaris/The National Key Facility for Crop Gene Resources and Genetic Improvement/Institute of Crop Sciences, Chinese Academy of Agricultural Sciences, Beijing 100081, China; haochenyang@caas.cn (C.H.); litian@caas.cn (T.L.)

**Keywords:** wheat, RNA-seq, DEGs, GO, stress treatment, RT-qPCR

## Abstract

Under climate change, drought is one of the most limiting factors that influences wheat (*Triticum aestivum* L.) production. Exploring stress-related genes is vital for wheat breeding. To identify genes related to the drought tolerance response, two common wheat cultivars, Zhengmai 366 (ZM366) and Chuanmai 42 (CM42), were selected based on their obvious difference in root length under 15% PEG-6000 treatment. The root length of the ZM366 cultivar was significantly longer than that of CM42. Stress-related genes were identified by RNA-seq in samples treated with 15% PEG-6000 for 7 days. In total, 11,083 differentially expressed genes (DEGs) and numerous single nucleotide polymorphisms (SNPs) and insertions/deletions (InDels) were identified. GO enrichment analysis revealed that the upregulated genes were mainly related to the response to water, acidic chemicals, oxygen-containing compounds, inorganic substances, and abiotic stimuli. Among the DEGs, the expression levels of 16 genes in ZM366 were higher than those in CM42 after the 15% PEG-6000 treatment based on RT-qPCR. Furthermore, EMS-induced mutants in Kronos (*T. turgidum* L.) of 4 representative DEGs possessed longer roots than the WT after the 15% PEG-6000 treatment. Altogether, the drought stress genes identified in this study represent useful gene resources for wheat breeding.

## 1. Introduction

As one of the most important Triticeae crops, common wheat (*Triticum aestivum* L.) has accompanied human civilizations for more than 10,000 years [1]. Wheat has a complex genome with a very large size and three subgenomes (AABBDD) [2]. Wheat production is seriously restricted by high salinity and drought [3]. With the continuous destruction of the environment, decreases in arable land, the continuous warming of the global climate, and reductions in water resources, the use of biological means to create and select superior drought-tolerant wheat varieties to solve the food crisis has become particularly important [4].

Salinity and drought are major constraints on crop growth and productivity, limiting sustainable agriculture in arid regions [5]. Drought tolerance has a significant impact on agriculture worldwide and is expected to continue to increase in the future [4]. Plants respond to drought tolerance via physiological, biochemical, morphological, and molecular mechanisms [4]. Understanding the physiology and molecular mechanisms of drought tolerance in crops is of vital importance for improving stress tolerance and promoting cultivation [4].

Compared with studies in model plants such as *Arabidopsis*, rice, and tomatoes, the application of traditional methods for mapping key genes and cultivating wheat varieties is very difficult. With the development of genome sequencing technology, omics methods have become an important approach for identifying key genes that encode important agricultural traits in wheat [6,7,8,9,10]. Among these methods, RNA-seq has been extensively utilized to map relevant genes, and it has become a commonly used and effective method for researchers. For example, the 15 calmodulin (*TaCAM*) and 113 calmodulin-like (*TaCML*) genes were identified based on the wheat reference genome, and the expression level of the *TaCAM2-D* gene was shown to be significantly upregulated under drought and salt stress [11]. To study the response of rice seedlings to high temperatures, RNA-seq was performed in two rice varieties, Annapurna (heat tolerant) and IR64 (heat sensitive); it revealed a unique set of regulatory genes and related pathways in the red rice variety Annapurna, including genes related to auxin and abscisic acid (ABA) in the rice heat-stress response [12]. *OsSPL10*, a gene regulating drought tolerance in rice, was cloned using forward genetics [13]. The molecular mechanism of an NF-Y-PYR module regulating drought and salt stress in soybeans was analyzed, and the results were of great significance for the development of tolerant soybean varieties [14]. An analysis of the transcriptome of developing wheat grains identified genes that can improve breeding efficiency and adaptation to climate change [15]. The drought-responsive gene *TaSNAC8-6A* was cloned by combining GWAS and RNA-seq techniques [16]. RNA-seq was also used to map the key gene *TaERF87*, which plays an important role in the PEG-6000-induced dehydration stress response [17]. Overall, drought stress has been studied in several crops, but there are only a few genes related to drought stress available in wheat, and further enrichment of such genetic resources is needed to adapt this crop to ecological changes in the environment.

Here, to study genes related to drought stress, RNA-seq analysis was used to identify differentially expressed genes (DEGs) in ZM366 and CM42 roots under 15% PEG-6000 treatment (to simulate drought), and the phenotypic differences between the two wheat cultivars were measured. Furthermore, the expression differences in 16 DEGs were analyzed among four Kronos mutants and the wild type. Overall, the results confirmed that these genes are novel stress-related genes of wheat that can be diligently used in a breeding process for drought-tolerant wheat cultivars; they will accelerate wheat molecular breeding for better crop production.

## 2. Results

### 2.1. Generation of RNA-Seq Data

Two ecologically differentiated wheat cultivars were selected based on their obvious difference in root length under 15% PEG-6000 treatment and wellwatered conditions (hydroponics) as a control. Estimation of root length and phenotypic analysis showed minor differences between the root length of the two cultivars under well-watered conditions; there was little difference in root growth between the two materials under hydroponics (Figure 1A,B and Figure 2A). However, the root length of the cultivar ZM366 was notably longer than that of CM42 under the 15% PEG-6000 treatment (Figure 1C,D and Figure 2A). In view of the above results, we believe that some important drought-tolerant genes are contained in ZM366. Furthermore, we collected the root tissues of the above plant material that had grown for seven days and proceeded with RNA-seq to explore the key drought-tolerant genes.

Twelve RNA-Seq libraries were constructed, with three biological repeats: ZM366_root_15%PEG_6000_1, ZM366_root_15%PEG_6000_2, ZM366_root_15%PEG_6000_3, CM42_root_15%PEG_6000_1, CM42_root_15%PEG_6000_2, CM42_root_15%PEG_6000_3, ZM366_root_hydroponics_1, ZM366_root_hydroponics_2, ZM366_root_hydroponics_3, CM42_root_hydroponics_1, CM42_root_hydroponics2, and CM42-root-hydroponics-3. We definitively compared ZM366 with CM42 under 15% PEG treatment. A total of 70.5–84.3 million reads were obtained from each sample. The amount of data ranged from 10.58 Gb to 12.64 Gb, the Q30 percentage exceeded 94.66%, and the GC content distribution was 55.93–57.28% (Appendix A). Sequencing alignment showed that 94.43–95.29% of the reads could be mapped to IWGSC RefSeq v2.1. The aligned regions, alignment rates, and sequencing data filtering were analyzed statistically. The regions identified in the sample comparisons were mainly exons, introns, and genes, with sample alignment rates of 85.08–85.89%, sequencing data filtering statistics that included clean data, low quality, containing N, and adapter related (Appendix A). The results suggested that the RNA-Seq data were suitable for further analysis.

### 2.2. Variant Site Detection and Analysis

The identified variant sites were mainly composed of SNPs and InDels. We analyzed ZM366 and CM42 under the 15% PEG treatment. The clean data obtained by sequencing were aligned with the wheat reference genome (version 2.1). For cultivar ZM366, the 3 biological repeats produced 82,003, 72,333, and 76,333 SNPs and 7934, 7190, and 7800 InDels. In CM42, the 3 biological repeats produced 73,554, 83,353, and 83,605 SNPs and 5856, 8189, and 6486 InDels (Appendix A). Next, the SNPs and InDels were annotated based on the Human Genome Variation Society (HGVS) annotation of variant sites at the DNA level (HGVS_C), variant sites at the protein level only for protein-encoding genes (HGVS_P), the effect of the mutation sites (EFFECT), and the degree of influence caused by the mutation sites (IMPACT) (Appendix A). The identification of SNPs and InDels has considerable implications for improving our understanding of the structural variation, genome evolution, and origin of wheat. Many SNPs and InDels were obtained through the analysis, which will aid in the later search for genes.

Next, statistics on each mutation site were performed based on the annotation information, including the function of the variant site, the mutation site area, the influence of the mutation site, and the detection of SNPs and InDels. The results showed that the effects of the mutation sites (SNP-impact and InDel-impact) were quantified and plotted at four levels: high, moderate, low, and modifier (no phenotypic effect itself, but effects on other mutation sites with phenotypic effects) (Figure 3A,D and Appendix A). The categories of the regions of variants (SNP-region and InDel-region) included the following: downstream, exon, intron, intergenic, splice_site_acceptor, splice_site_donor, splice_site_region, transcript, upstream, UTR_3_prime, and UTR_5_prime (Figure 3B,E). The variant site function (SNP-function) varied from synonymous mutations to missense mutations and nonsense mutations (Figure 3C). Finally, after we obtained the SNP sites, we clarified the functions of these mutation sites, the regions of the mutations, and the impacts of the mutation sites. These characteristics may play a role in predicting the effect of genetic mutations on gene function. By counting the mutation sites, we also develop a clearer understanding of the changes in key genes; this will help to identify key genes and predict upstream and downstream genes that regulate their expression.

### 2.3. Principle Component Analysis (PCA)

Principle Component Analysis (PCA) was performed to evaluate inter-group differences and intra-group sample duplication. It is an algorithm to reduce the dimensions of data. The distance between points in the figure represents the degree of similarity between samples. PC1 refers to the top contribution rate, which is the factor that has the greatest influence on variation, and PC2 is the second factor. Therefore, we tend to pay more attention to the distribution of samples on PC1 (the horizontal coordinate). To assess inter-group differences and sample duplication within groups, PCA analysis was performed for the gene expression values of all samples. As shown in Figure 4, the PC1 and PC2 of the three samples of CM42-hydroponics, CM42-15% PEG, ZM366-hydroponics, and ZM366-15% PEG group are very close, and such samples are actually very similar. Figure 4A represents a PCA analysis of a PCA 2D plot. Figure 4B represents a PCA analysis of a PCA 3D plot. Overall, sample duplication within the group was consistent, and the PCA data showed the high quality of the samples’ biological repetition.

### 2.4. Identification of Differentially Expressed Genes

Gene expression quantification analysis was performed, and the expected number of fragments per kilobase of transcript sequence per million base pairs sequenced (FPKM) was calculated. The read count expression matrix and the FPKM expression matrix of each gene in the sample are shown in Appendix A. Based on the gene quantification results, 49,216 genes were expressed in both ZM366 and CM42, and 4426 and 3726 genes were uniquely expressed in either ZM366 or CM42, respectively (Figure 5A, Appendix A). DEG analysis showed that there were 11,083 DEGs between ZM366 and CM42 after 15% PEG-6000. Among the DEGs, 6329 genes were upregulated, and 4844 genes were downregulated (Figure 5B, Appendix A). In addition, we studied the DEGs of ZM366 and CM42 under control conditions (hydroponics), and there were 13,235 DEGs between ZM366 and CM42. Among the DEGs, 6937 genes were upregulated, and 6298 genes were downregulated (Appendix A). Among these DEGs, 7294 DEGs were uniquely identified under PEG-6000 treatment conditions. A total of 4111 genes were upregulated, and 3183 genes were downregulated (Appendix A). 

### 2.5. GO Enrichment Analysis of Differentially Expressed Genes (DEGs)

After obtaining the DEGs, we focused on the functions of the genes and further analyzed the associated biological pathways. The Gene Ontology (GO) database is a comprehensive database describing gene functions, which are divided into three major categories: biological process (BP), cellular component (CC), and molecular function (MF). GO enrichment analysis revealed 2582 significantly enriched genes, including 1293 upregulated genes and 1289 downregulated genes (Appendix A). The 1293 upregulated genes were further analyzed. Based on the analysis, the most prominent BP terms were the response to water, acid chemicals, oxygen-containing compounds, inorganic substances, and abiotic stimuli. The result is shown by GO enrichment analysis bubble diagrams (Figure 5C). Importantly, a total of 41 shared genes were annotated with these terms, and gene expression analysis showed that the expression heatmap of these 41 genes was higher in ZM366 than in CM42 (Figure 5D). Next, we focused on 16 of the 41 genes associated with these terms. These 16 genes were the most significantly enriched in the GO pathways; based on their association with the same enrichment pathway, we considered these genes to be potential candidates related to drought stress (Table 1).

### 2.6. Quantitative Real-Time PCR Analysis and Functional Verification of Candidate Genes

To further identify candidate genes, RT-qPCR analysis was completed. On day 7 after ZM366 and CM42 were treated with a 15% PEG-6000 solution, root tissues were collected, and RNA was extracted. RT-qPCR showed that the expression levels of 16 genes in ZM366 were higher than those of the genes in CM42 (Figure 6, Appendix A). Therefore, these 16 genes were regarded as important candidate genes.

To verify that the candidate genes were indeed related to drought stress tolerance, we checked the tetraploid wheat Kronos (*Triticum turgidum* L.) EMS-induced mutant information for these genes in the Ensembl plant database (http://plants.ensembl.org/Triticum_aestivum/Info/Index, accessed on 25 August 2022) and found available mutant information for four genes. These gene ID were TraesCS6A03G0900700 (TraesCS6A02G350500), TraesCS6A03G0899800 (TraesCS6A02G350100), TraesCS3B03G1055400 (TraesCS3B02G428200), and TraesCS6B03G1084600 (TraesCS6B02G383800), and the respective EMS-induced mutant identifier numbers were Kronos3935.chr6A.582265082, Kronos3216.chr6A.581983177, Kronos3538.chr3B.667112943, and Kronos3557.chr6B.658578115 (abbreviated as 3935, 3216, 3538, and 3557); the tetraploid wheat Kronos was used as the control. The mutants’ information is listed in Table 2. 

Next, the 4 EMS-induced mutants seeds were planted and treated with 15% PEG-6000 solution, and root growth was observed at 2, 4, 5, and 7 days after planting. Based on phenotypic characteristics and root growth statistics analysis, we found that the root growth rate of the mutants was apparently faster, and the roots length were longer than the control (Figure 2B and Figure 7). This experiment was repeated three times with similar results. 

In addition, further verification was conducted using RT-qPCR with three biological repeats for each group. The expression levels of the corresponding genes in the four mutant materials were significantly higher than those in the control group Kronos material (Figure 8, Appendix A), which is consistent with the result that the mutants’ roots were longer than WT. Therefore, it is reasonable to assume that these genes are involved in drought tolerance pathways in wheat. Therefore, these genes can be regarded as important candidate genes related to drought stress.

## 3. Discussion

### 3.1. Importance of Drought Resistance Genes in Wheat According to Multi-Omics

Global climate change gives rise to environmental cues, especially drought tolerance [4]. In this study, 16 drought tolerance genes with high confidence level were obtained by RNA-seq, and functional verification of the candidate genes was carried out. This provides data in support of the study of drought tolerance genes and plays a supporting role in enriching wheat’s genetic resources for drought tolerance.

Additionally, drought stress disturbs seed germination, seedling development, leaf size, plant height, and plant biomass, limiting water availability to roots and leading to losses in crop yield [17]. Deeper and longer roots can help crops to access soil water under water-deficient conditions, increasing production [4,18]. In general, plants are in a state of drought tolerance when water transfer to the root system is insufficient, or when the amount of water lost through transpiration is very high [19]. Damage caused by drought tolerance severity is often unstable because it is determined by several factors, such as rainfall patterns, soil moisture availability, and water deficiency due to transpiration [20]. As a result, drought tolerance hinders crop growth, water-nutrient relationships, and photosynthesis, ultimately leading to a significant decline in crop yields [4,19,20]. Therefore, studying wheat drought resistance is an important and meaningful aspect of wheat breeding. Multi-omics analysis has become a mainstream method for mapping important agronomic loci of wheat. Notably, at present, the development of RNA-seq, genomics-assisted breeding, proteomics, metabolomics, epigenetics, pan-genomics and high-throughput phenomics provides a convenient approach for the exploration of important agronomic trait resources and the cloning of important agronomic genes in wheat, which helps us understand drought tolerance adaptation and molecular mechanisms [4]. 

Recently, scientists have used genome-wide association studies (GWASs), RNA-seq, and DAP-seq to clone the wheat drought tolerance genes *TaDTG6-B* [21], *TaNAC071-A* [22], and *TaWRKY1-2D* [23]. Using two control genotypes, drought-tolerant NI543941 and drought-susceptible WL711, a total of 45,139 DEGs, 13,820 TFs, 288 miRNAs, and 640 pathways were identified to provide an RNA-Seq approach for studying the response mechanism of wheat roots under drought conditions [24]. However, none of these new genes that have been reported so far were included in the current study. Therefore, the new genes found via RNA-Seq in this study represent a significant development for wheat drought resistance breeding and the cloning of new drought resistance genes.

### 3.2. Sixteen Key Candidate Drought Tolerance Genes Identified in the Response to 15% PEG-6000 Treatment

In previous studies, the researchers found that *TaERF87* and *TaAKS1* synergistically regulate *TaP5CS1*/*TAP*5Cr1-mediated proline biosynthesis to enhance drought resistance in wheat [25]. Some researchers explored new QTL loci in response to drought stress [26]. *TaIAA15-1A* and *TaSPP-5A* were also associated with drought tolerance [27,28]. These key genes, associated with tolerance to drought stress, have been cloned and excavated using various methods. However, we found that the new genes obtained in this study are not the same type of gene as was previously reported. 

In this study, 16 key candidate genes responding to drought stress were identified by RNA-seq, RT-qPCR, and tetraploid wheat mutants. These genes showed the most significant results in the GO enrichment pathway, and their expression levels were also highest in the same enrichment pathway. Through our analysis, we found an interesting phenomenon: all 16 of these genes belong to the dehydrin family. Numerous reports have shown that the dehydrin gene is involved in regulating drought resistance in plants. The dehydrin gene exhibits specialized metabolite drought stress responses in switchgrass [29]. The dehydrin genes *Gh_A05G1554* (*GhDHN_03*) and *Gh_D05G1729* (*GhDHN_04*) enhanced cotton drought tolerance and salt tolerance [30]. Dehydrin *MtCAS31* promotes autophagic degradation under drought stress [31]. The researchers studied the regulatory mechanism of the wheat dehydrin gene *WZY2* under drought stress [32]. Therefore, it is reasonable to assume that these genes have important application value for wheat drought resistance breeding and play an important role in the wheat drought resistance pathway. This study provides a reference for the future enrichment of key gene resources and cloning of genes related to drought stress tolerance. 

### 3.3. Genomic Era Genome Data Integration and Establishment of a Stress Gene Resource Exploration Platform

In this experiment, a large number of reliable SNPs and InDels were identified. Therefore, it is reasonable to assume that these variations may be one of the reasons why ZM366 is more drought tolerant than CM42. It can be used to explain the reason for the significant differences in genotype and phenotype between the two cultivars. We can better understand the sequence and structural variation between the two phenotypically different cultivars. These data assist the mapping of the corresponding genes related to drought resistance, providing support for the later utilization and screening of excellent alleles and cultivars. With the explosive growth of information and data, quickly mapping valuable target genes and loci has become a research hotspot in genomics [33]. Thus, there is currently increased interest in the fast and accurate anchoring of key regions and candidate genes. The integration and utilization of massive amounts of data to establish a stress gene resource exploration platform based on high-throughput genotype identification has become extremely urgent [34]. To date, a large number of gene resource exploration platforms [33,34,35,36,37], wheat databases [38,39,40], and websites [41] have been built, enabling researchers to consult genomic information more efficiently.

There is no doubt that our next step is to clone and use these key genes. Transgenics, gene editing, and other technologies can be used to create excellent wheat varieties. The cloning and utilization of these key genes provide key data support for the breeding of drought-tolerant wheat varieties and the creation of drought-tolerant cultivars, and they will enrich the drought-tolerant gene pool of wheat. We will build and improve the database, using existing data and databases to discover and explore differences in sequence changes between modern breeding varieties and local cultivars to quickly search for superior alleles and superior key genes. Additionally, our work will provide ideas for the creation of excellent drought-tolerant germplasm resources and materials through modern and advanced biological techniques. Under the conditions of global warming and the decrease in the effective use of cultivated areas, there is a demand for agricultural development to identify key genes and create planting materials using biotechnology and database platforms. Thus, speed breeding can be realized in the near future.

## 4. Materials and Methods

### 4.1. Materials and Growth Conditions

To identify genes related to the drought tolerance response, two different ecologically differentiated materials, Zhengmai 366 (ZM366, Wheat Research Institute of Henan Academy of Agricultural Sciences, Henan, China) and Chuanmai 42 (CM42, Crop Research Institute of Sichuan Academy of Agricultural Sciences, Sichuan, China), were treated with 15% polyethylene glycol (PEG-6000, Modern Oriental Technology Development CD., LTD, Beijing, China) for 7 days to observe root growth changes; a water treatment was used as a control. Each treatment was repeated three times. Moreover, 4 EMS-induced mutants of Kronos (*Triticum turgidum* L.) (Table 2) were selected to be treated with 15% PEG-6000 for 7 days to observe root growth changes, using Kronos as a control. Each sample treatment was repeated three times.

### 4.2. RNA Extraction, Library Construction, Sequencing, and Quality Control

In this experiment, RNA samples were stored and sequenced after 7 days of growth. Total RNA was extracted using standard methods, followed by strict quality inspection and the on-machine sequencing of RNA sequencing data [42]. The constructed sequence libraries subjected to 150 bp paired-end sequencing on the Illumina HiSeq™ 6000 platform, and the raw data were saved as FASTQ files. To remove adaptors and low-quality reads at the head and tail, Trimmomatic (version 0.35) [43] was used to conduct quality control of the sequencing data using the following protocol: java–jar, trimmomatic-0.35.jar, PE-threads 20, forward.fastq, reverse.fastq, forward_paired.fastq, forward_unpaired.fastq, reverse_paired.fastq, reverse_unpaired.fastq, ILLUMINACLIP:1.adapter.list:2:30:10, LEADING:10, TRAILING:10, SLIDINGWINDOW:1:10 and MINLEN:50. In this step, clean data were obtained by removing reads containing adapters, reads containing poly-N, and low-quality reads from the raw data. At the same time, the Q20, Q30, and GC contents of the clean data were calculated. FaQC software (version 2.08) [44] was then used to calculate the quality score, Q20, Q30 distribution, and GC distribution using the default parameters: FaQCs—1 R1.clean.fastq—2 R2.clean.fastq—d.—qc_only. PCA of RNA−seq data was carried out by OmicShare tools (http://www.omicshare.com/tools, accessed on 15 September 2021). All downstream analyses were performed based on high-quality clean data.

### 4.3. Read Mapping to the Reference Genome and Novel Transcript Prediction

The wheat reference genome (2.1 version) and gene model annotation files were directly downloaded from the genome website (IWGSC RefSeq v2.1, http://www.wheatgenome.org/Tools-and-Resources, accessed on 10 August 2021). The index of the reference genome was built using HISAT2 (v2.0.5), and paired-end clean reads were aligned to the reference genome using Hisat2 (v2.0.5). Clean reads were mapped to IWGSC RefSeq v2.1 by HISAT2 with the parameters “hisat2—x reference.genome.index—p 8—X 400—no-unal—dta—1 input.R1.clean.fastq.gz—2 input.R2.clean.fastq.gz—S input.sam”, and the mapping results of the reads were stored in a BAM file [45]. The mapped reads of each sample were assembled using StringTie (vl.3.3b) using a reference-based approach [46]. StringTie uses a novel network flow algorithm as well as an optional de novo assembly step to assemble and quantitate full-length transcripts representing multiple splice variants for each gene locus.

### 4.4. SNP Analysis

GATK [47] (v4.1.1.0) software was used to perform SNP calling. Raw VCF files were filtered with the GATK standard filter method and the following parameters settings: cluster, 3; window size, 35; QD < 2.0; FS > 30.0; and DP < 10. SnpEff (v4.3.1) software was used for variant site annotation [48].

### 4.5. Quantification of Gene Expression Levels

FeatureCounts (vl.5.0-p3) [49] was used to count the number of reads mapped to each gene. The FPKM value of each gene was then calculated based on the length of the gene and the read count mapped to the gene. The FPKM value, representing the expected number of fragments per kilobase of transcript sequence per millions of base pairs sequenced, simultaneously considers the effect of the sequencing depth and gene length on the read count; it is currently the most commonly used metric for estimating gene expression levels [50,51].

### 4.6. Differential Expression Analysis (DEGs)

The differential expression analysis of three biological repeats per condition was performed using the DESeq2 R package (v1.20.0) [51]. DESeq2 provides statistical routines for determining differential expression from digital gene expression data using a model based on the negative binomial distribution. The resulting *p* values were adjusted using Benjamini and Hochberg’s approach for controlling the false discovery rate. Genes identified by DESeq2 with an adjusted *p* value < 0.05 were assigned as differentially expressed.

### 4.7. GO Enrichment Analysis of Differentially Expressed Genes

The clusterProfiler R package was used for the Gene Ontology (GO) enrichment analysis of differentially expressed genes [52], in which gene length bias was corrected. GO terms with corrected *p* values less than 0.05 were considered significantly enriched by differentially expressed genes.

### 4.8. Quantitative Real-Time PCR

NCBI-Primer-BLAST (https://www.ncbi.nlm.nih.gov/tools/primer-blast, accessed on 15 November 2022) was used to design gene-specific primers. Each experiment included 3 biological repeats. In the PCR system, each 20 µL reaction contained 10 µL of SYBR Mix, 2 µL of cDNA, 0.8 µL of forward and reverse primers, and 7.2 µL of ddH2O. RT-PCR amplification was performed in a Roche LightCycler 480 Real-Time System (Roche, Switzerland). The PCR amplification steps were 95 °C for 60 s and then 40 cycles of 95 °C for 5 s, 60 °C for 30 s, and 95 °C for 15 s. The 2 (-delta C (T)) method was used for the relative gene expression analysis of DEGs with TaActin as the endogenous control [53]. Data are presented as the mean ± standard deviation.

## 5. Conclusions

This study revealed that ZM366 was more drought resistant than CM42; 16 key genes that were responsive to drought stress were identified by RNA-seq and RT-qPCR, and 4 genes related to drought resistance were further functionally verified by mutant analysis and RT-qPCR. This work provides data that will support studies on drought stress in wheat and the mapping of wheat drought resistance genes, as well as the enrichment of wheat drought genetic resources. In addition, a large number of SNPs and InDels were identified in this study, which will contribute to the study of the origin, evolution, and genetic diversity of wheat in the postgenomic era and will play an important role in enriching wheat databases.

The next focus of our work will be to systematically study the 16 genes obtained in this work. We will clone the key genes, verify their functions through overexpression, gene editing, and other methods, and analyze the mechanism of the specific regulation of drought tolerance by these genes via biochemical experiments.

## Figures and Tables

**Figure 1 ijms-24-07245-f001:**
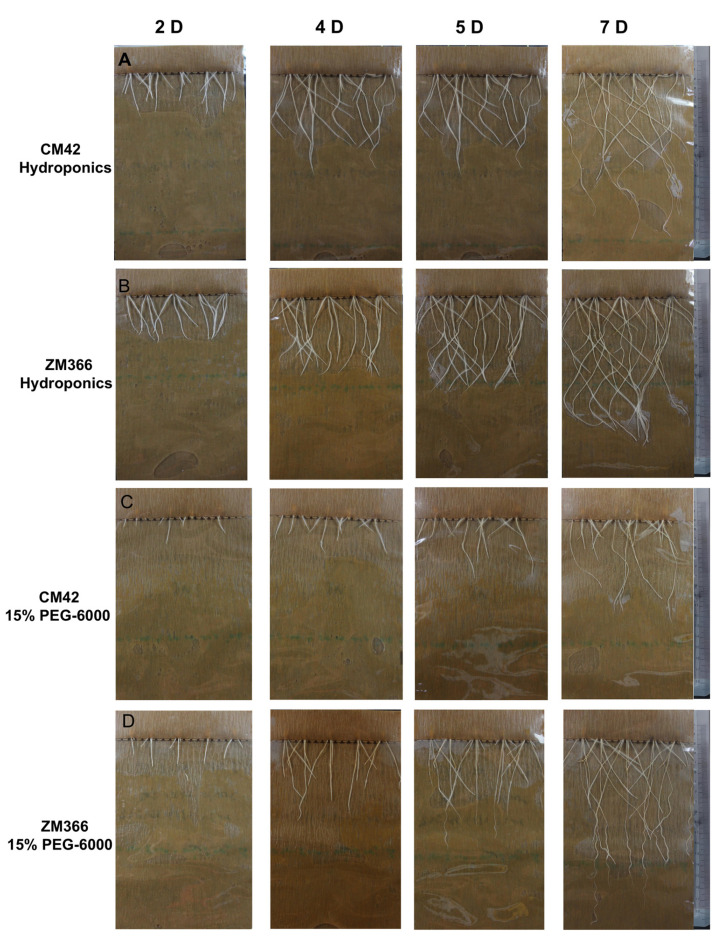
Root growth of CM42 and ZM366. (**A**) CM42 root growth under hydroponic conditions. (**B**) ZM366 root growth under hydroponic conditions. (**C**) CM42 root growth under the 15% PEG-6000 treatment. (**D**) ZM366 root growth under the 15% PEG-6000 treatment. Root growth was measured on days 2, 4, 5, and 7.

**Figure 2 ijms-24-07245-f002:**
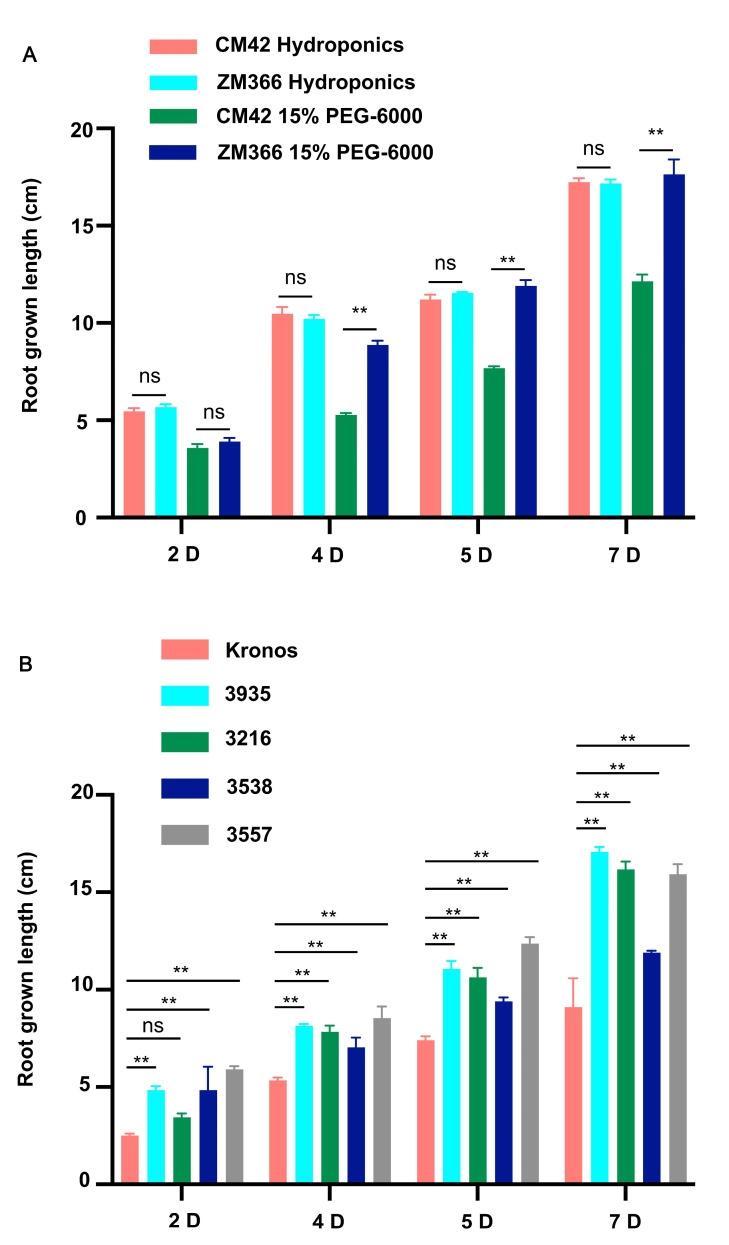
Root growth length statistics. (**A**) Statistics of CM42 and ZM366 under hydroponics and the 15% PEG-6000 treatment. (**B**) Statistics of Kronos and 4 EMS-induced mutants under the 15% PEG-6000 treatment. Growth was measured on days 2, 4, 5, and 7. The experiment was repeated three times, and the error bar represents the SD of the means (*n* = 3). ** represents *p* < 0.01. ns indicates not significant.

**Figure 3 ijms-24-07245-f003:**
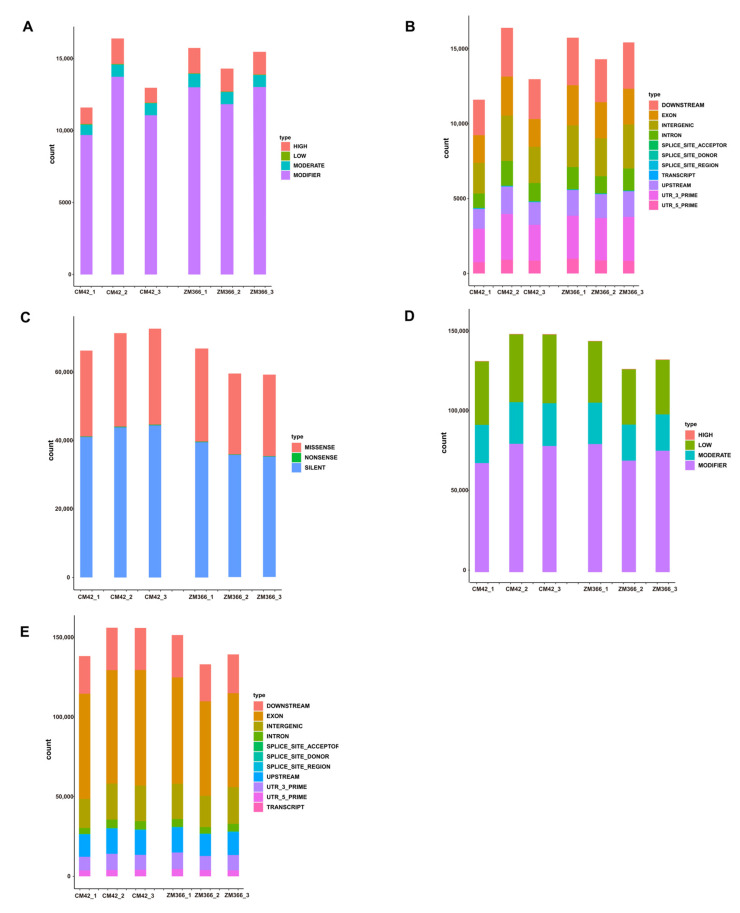
Statistics for variant sites, including InDels and SNPs. (**A**) InDel-impact was plotted for four levels: high, moderate, low, and modifier. (**B**) InDel-region was counted for the following gene structure regions: downstream, exon, intron, intergenic, splice_site_acceptor, splice_site_donor, and splice_site_region, transcript, upstream, utr_3_prime and utr_5_prime. (**C**) SNP-function was statistically plotted for three conditions: synonymous mutation, missense mutation, and nonsense mutation. (**D**) SNP-impact was statistically plotted for four levels: high, moderate, low, and modifier. (**E**) SNP-region was counted and mapped for the following gene structure regions: downstream, exon, intron, intergenic, splice_site_acceptor, splice_site_donor, and splice_site_region, transcript, upstream, utr_3_prime and utr_5_prime.

**Figure 4 ijms-24-07245-f004:**
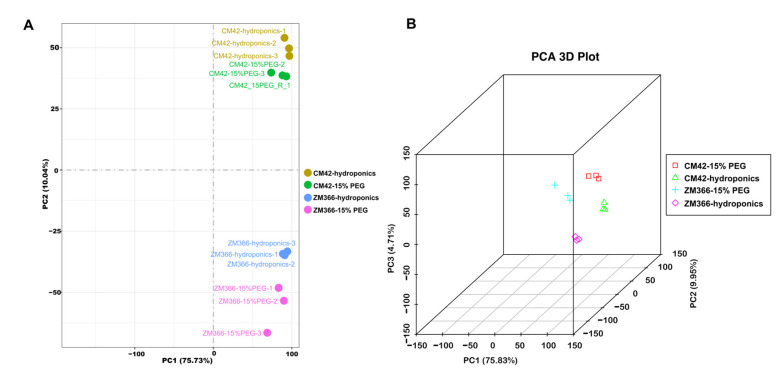
Principle Component Analysis (PCA) (**A**) CM42-hydroponics, CM42-15% PEG, ZM366-hydroponics, and ZM366-15% PEG PCA of a PCA 2D plot. (**B**) CM42-hydroponics, CM42-15% PEG, ZM366-hydroponics, and ZM366-15% PEG PCA of a PCA 3D plot. PC1 refers to the top contribution rate, which is the factor that has the greatest influence on variation, and PC2 is the second factor.

**Figure 5 ijms-24-07245-f005:**
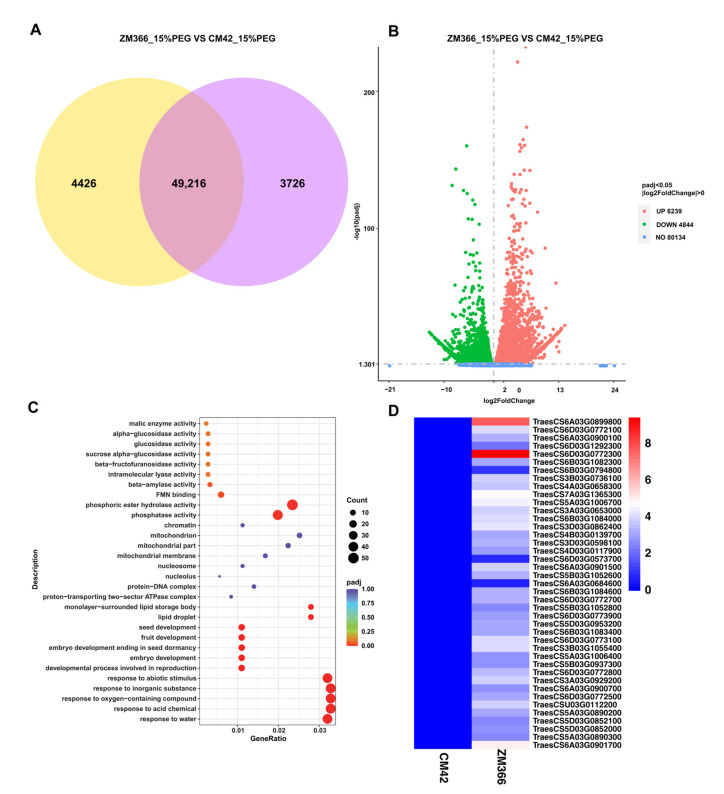
Transcriptomic analysis of ZM366 and CM42 treated with 15% PEG-6000. (**A**) Venn diagram of expressed genes in ZM366 and CM42 under 15% PEG-6000 treatment conditions. The yellow represents the genes expressed in ZM366. The purple represents the genes expressed in CM42. (**B**) Volcano map of differentially expressed genes (DEGs) in ZM366 and CM42 under 15% PEG-6000 treatment conditions. Up, 6239; down, 4844. (**C**) GO enrichment analysis bubble diagrams. The abscissa represents the ratio of the number of DEGs annotated with a GO term to the total number of DEGs, and the ordinate is the GO term; the size of the point represents the number of genes annotated with a GO term, and the color change from red to purple represents the significance of the enrichment. (**D**) Expression heatmap of the GO enrichment DEGs between CM42 and ZM366 under 15% PEG-6000 treatment conditions.

**Figure 6 ijms-24-07245-f006:**
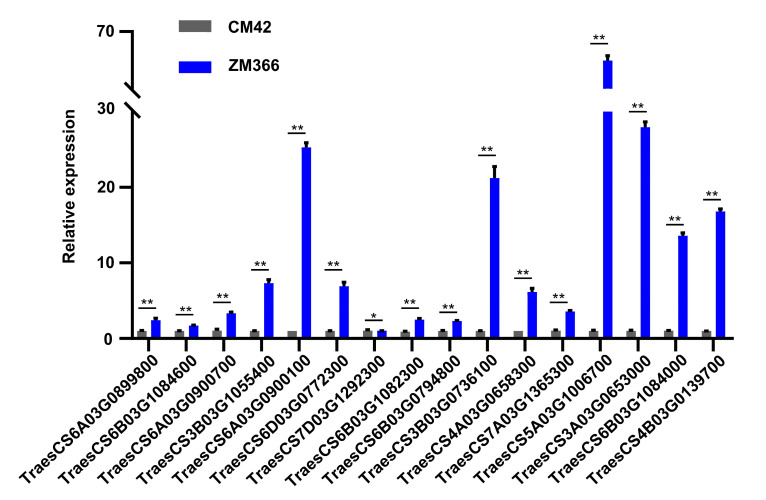
Expression patterns revealed by RT-qPCR. The expression patterns of 16 highly expressed genes in ZM366 compared with CM42 (both treated with 15% PEG-6000) as revealed by RT-qPCR. The experiment was repeated three times, and the error bar represents the SD of the means (*n* = 3). ** represents *p* < 0.01. * represents *p* < 0.05.

**Figure 7 ijms-24-07245-f007:**
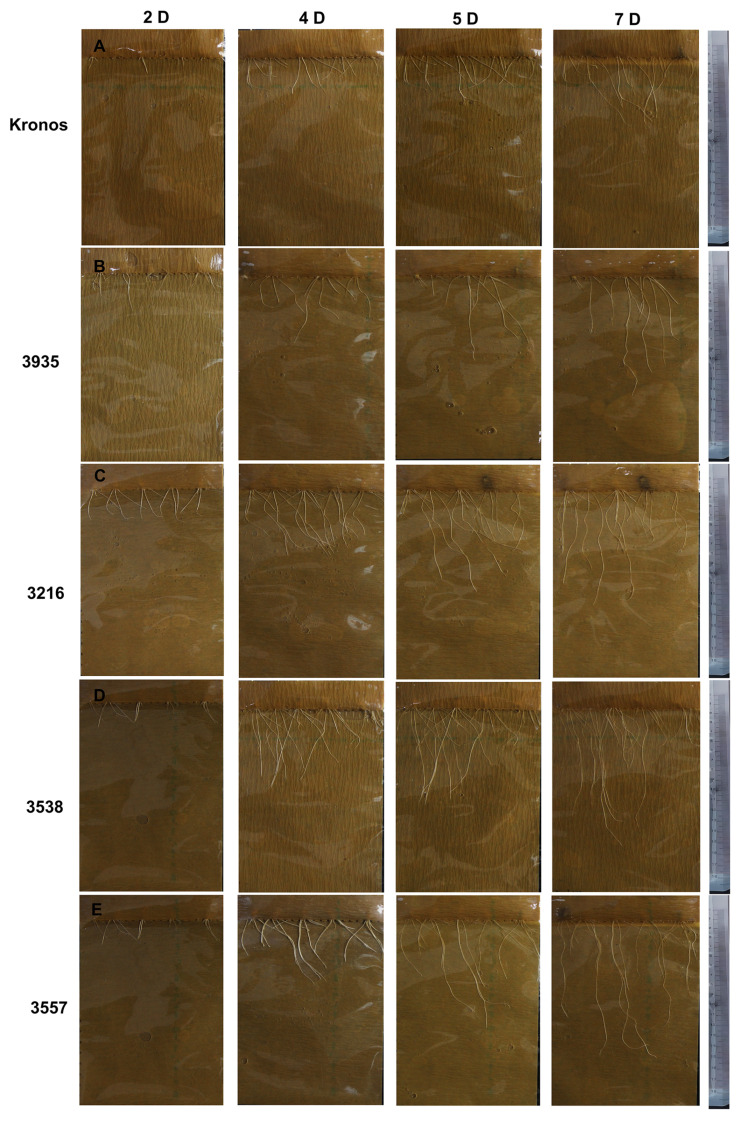
Root growth of Kronos and 4 EMS-induced mutants under the 15% PEG-6000 treatment. (**A**) Kronos root growth conditions. (**B**) Root growth conditions of mutant 3935. (**C**) Root growth conditions of mutant 3216. (**D**) Root growth conditions of mutant 3538. (**E**) Root growth conditions of mutant 3557. Root growth was measured on days 2, 4, 5, and 7.

**Figure 8 ijms-24-07245-f008:**
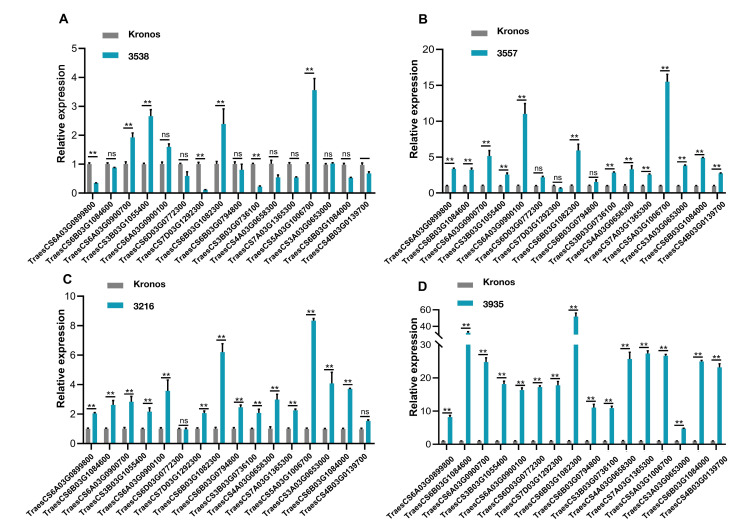
Expression patterns revealed by RT-qPCR treated with 15% PEG-6000. (**A**) The expression patterns of 16 highly expressed genes in mutant 3538 compared with Kronos, mutant 3557 (**B**), in mutant 3216 (**C**), and in mutant 3935 (**D**). The experiment was repeated three times, and the error bar represents the SD of the means (*n* = 3). ** represents *p* < 0.01. ns indicates not significant.

**Table 1 ijms-24-07245-t001:** Information for the 16 candidate genes.

Candidate Gene ID	Description	Log2(FC)	*p* Value
TraesCS6A03G0899800	Dehydrin, BP: response to water, abiotic stimulus	7.60	2.92 × 10^−45^
TraesCS6B03G1084600	Dehydrin, BP: response to water, abiotic stimulus	3.40	1.47 × 10^−41^
TraesCS6A03G0900700	Dehydrin, BP: response to water, acid chemical	2.77	2.92 × 10^−45^
TraesCS3B03G1055400	Dehydrin, BP: response to water, abscisic acid	4.12	2.92 × 10^−45^
TraesCS6A03G0900100	Dehydrin, BP: response to water, abscisic acid	3.22	1.38 × 10^−27^
TraesCS6D03G0772300	Dehydrin, BP: response to water, inorganic substance	2.24	5.15 × 10^−17^
TraesCS7D03G1292300	Dehydrin, BP: response to water, oxygen-containing compound	9.46	8.39 × 10^−15^
TraesCS6B03G1082300	Dehydrin, BP: response to water, inorganic substance	3.13	1.23 × 10^−12^
TraesCS6B03G0794800	Dehydrin, BP: response to water	0.99	1.80 × 10^−37^
TraesCS3B03G0736100	Dehydrin, BP: response to water, abiotic stimulus	3.92	2.19 × 10^−36^
TraesCS4A03G0658300	Dehydrin, BP: response to water	3.76	4.84 × 10^−8^
TraesCS7A03G1365300	Dehydrin, BP: response to water, abiotic stimulus	4.81	1.47 × 10^−41^
TraesCS5A03G1006700	Dehydrin, BP: response to water, oxygen-containing compound	4.47	3.94 × 10^−7^
TraesCS3A03G0653000	Dehydrin, BP: response to water, abscisic acid	3.94	3.96 × 10^−7^
TraesCS6B03G1084000	Dehydrin, BP: response to water, inorganic substance	4.00	1.8 × 10^−37^
TraesCS4B03G0139700	Dehydrin, BP: response to water, abiotic stimulus	3.12	2.19 × 10^−36^

**Table 2 ijms-24-07245-t002:** EMS-induced mutation information.

Mutation Name	Most Severe Consequence	Alleles	Location of IWGSC RefSeq v2.0	IWGSC RefSeq v2.1 (IWGSC RefSeq v2.0)
Kronos3216.chr6A.581983177	splice region variant	G/A	Chromosome 6A:581983177	TraesCS6A03G0899800 (TraesCS6A02G350100)
Kronos3935.chr6A.582265082	splice region variant	C/T	Chromosome 6A:582265082	TraesCS6A03G0900700 (TraesCS6A02G350500)
Kronos3538.chr3B.667112943	splice region variant	G/A	Chromosome 3B:667112943	TraesCS3B03G1055400 (TraesCS3B02G428200)
Kronos3557.chr6B.658578115	splice region variant	C/T	Chromosome 6B:658578115	TraesCS6B03G1084600 (TraesCS6B02G383800)

## Data Availability

The datasets used and analyzed during the current study have been successfully stored in the SRA database of NCBI; the RNA-Seq accession number is PRJNA950485. The reference genome and annotation files of hexaploid wheat Chines Spring (IWGSC RefSeq v2.1) were downloaded from the http://www.wheatgenome.org/Tools-and-Resources (accessed on 10 August 2021).

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
