# Peer review of "Transcriptome Analysis of Roots from Wheat (Triticum aestivum L.) Varieties in Response to Drought Stress"

_ijms, 2023, doi:10.3390/ijms24087245_

Round 1

Reviewer 1 Report

In this study, transcriptome analysis between ZM366 and CM42 under PEG-6000 treatment condition were performed. ZM366 was proved to be more drought resistant than CM42, and 16 key genes responding to drought stress were identified by RNA-seq and RT-qPCR. Four key DEGs were functionally verified by mutants for their involvement in drought tolerance response. This is an useful and meaningful study. I suggest to accept it for publication after revision. My major concerns are as follows:

1. The genus name in Latin, such as “Triticum aestivum L.” in Line 16, “T. turgidum L.” in Line 26, needs to be italicized.

2. Lines 102-105: For cultivar CM42, three biological repeats produced 5856, 8189, and 6486 InDels and 73554, 83353, and 83605 SNPs. In ZM366, three biological repeats produced 7394, 7190, and 7801 InDels and 82003, 72333, and 76333 SNPs (Table S4). I cannot understand the origin of these DNA sequence variation among the same cultivar in three different biological samples. Please rephrase the corresponding description to make it easily be understood. In addition, please check and correct the numeral in the main text according to the corresponding supplementary materials.

3. Only differential expression analysis of ZM366 and CM42 treated with 15% PEG-6000 was performed. How is about the DEG between ZM366 and CM42 under control condition? How many DEG were uniquely identified under PEG-6000 treatment condition? Please supplement the corresponding information, if possible.These unique DEG under PEG-6000 treatment between ZM366 and CM42 should be of great value in understanding the genetic regulatory mechanism of drought tolerance/response.

4. There are some grammatical errors in this article. Please correct them carefully.

Reviewer 2 Report

Xi et al submitted a manuscript titled "Transcriptome analysis of roots from wheat (Triticum aestivum L.) varieties in response to drought stress". It is well written with some interesting results. This qualifies for publication as such in IJMS.

Reviewer 3 Report

The paper by Xi et al. entitled 'Transcriptome analysis of roots from wheat (Triticum aestivum L.) varieties in response to drought stress' is dealing with an important topic which is exploring stress-related genes of whear root particularly important under the ongoing climate change context and global warming.

1/Abstract section is fine and clear

2/The authors missed to insert the introduction section since this part contains the guidelines generally found in the template.

3/Materials and methods section is good and clear

4/Results: in this section authors made several mistakes too many figures

Misatkes in numbering the figures and their citation within the text

Duplicated figures titles

Try to avoid using too much figures besides the supplemetary files  

5/Discussion I feel personally that this is poor and needs to be reformulated concentrating on the newly obtained results with respect to previous works

6/the Conclusion section lacks 1-2 sentences regarding where future studies should preferably focus based on the present work

Many suggestions are directly inserted in attached file

Reviewer 4 Report

Overall, the study is worthy but the current draft is poorly organized, and some information is missing. Thus, I recommend major revision so that authors can resubmit a modified version for re-review.

Scientific names, including the title, should always be in italics throughout the manuscript.

Throughout the manuscript, please change resistance/resistant with tolerance/tolerant.

Where is introduction text? Most probably, authors forget to add an introduction. Please add an introduction in the next version. The introduction and discussion need to be supported by recent studies (doi: /10.1002/tpg2.20279), and explain how drought stress impacts agricultural production.

In results, please also explain what the data reflects. Do not simply explain the figures.

Add the PCA figure in the main file.

All figures are poorly arranged. Please remove the figure's legends which are part of the image itself. Also, these legends are not consistent with the main legends. Please rework the figures and their legends. Also, remove the extra spaces within the images.

There are several writing errors in the whole manuscript.

Have you submitted the RNA-seq data to any public repository?

Discussion is too short and lacks mechanistic insights.

Overall, the current needs to be significantly improved.

Round 2

Reviewer 3 Report

The authors have well revised the manuscript 

I would suggest to accept it 

Reviewer 4 Report

The revised version can be accepted for publication. Some minor writing errors could be corrected during proofing.